# Geometry-Preserving Unsupervised Alignment for Heterogeneous Foundation Models

**Shuwen Yu** [1]  **Zhanxuan Hu** [1]  **Yi Zhao** [1]  **Yonghang Tai** [1]  **Huafeng Li** [2]

## Abstract

Foundation models have driven rapid progress in computer vision, yet the two dominant paradigms, vision-language foundation models (VLMs) and vision-only foundation models (VFMs), remain only partially compatible. VLMs offer language-grounded semantic alignment but are often visually coarse, while VFMs learn discriminative perceptual geometry but lack semantic grounding. We propose **GPUA**, a *Geometry-Preserving Unsupervised Alignment* framework that integrates the complementary strengths of VFMs and VLMs. Inspired by cross-lingual alignment, **GPUA** treats VFM features as a *visual language* and learns an *orthogonal* mapping that translates the VFM space into the VLM semantic space, preserving geometry and narrowing the modality gap *without labels and model parameter updates*. **GPUA** is task-agnostic and requires only feature-level access to pretrained models. Experiments across diverse benchmarks demonstrate improved cross-model compatibility and strong gains in downstream zero-shot recognition and segmentation with negligible overhead. Our code is available at: https://github.com/Yuteam14/GPUA.

## 1. Introduction

Foundation models have become the cornerstone of modern computer vision, where two paradigms dominate: vision–language foundation models (VLMs) and vision-only foundation models (VFMs). VLMs, exemplified by CLIP (Radford et al., 2021), provide a powerful language-grounded semantic interface that enables open-vocabulary recognition

and strong cross-domain transfer. However, their visual representations are often semantically aligned yet perceptually coarse, making them less sensitive to fine-grained structures and local details. In contrast, VFMs such as DINO-style self-supervised models (Oquab et al., 2023; Siméoni et al., 2025) learn highly discriminative visual representations with strong locality and structural awareness, but lack explicit semantic grounding and struggle to support open-vocabulary reasoning. These complementary strengths and limitations naturally raise an important question:

> *Can we integrate heterogeneous foundation models to obtain representations that are both semantically grounded and perceptually discriminative?*

Recent efforts on multi-foundation model fusion seek to exploit the complementary strengths of heterogeneous pretrained models, particularly in open-vocabulary semantic segmentation (Dong et al., 2023). A common paradigm uses CLIP as the source of open-vocabulary semantics, augments it with DINO-style VFMs to enhance discriminative patch-level cues (Wysoczańska et al., 2024; Barsellotti et al., 2025), or further leverages SAM-style promptable segmenters for high-quality mask generation (Yang & Gong, 2024; Sun et al., 2024; Zhang et al., 2025). Through patch-level prediction and cross-model consistency, such pipelines partially compensate for CLIP's limitations in localization and fine-grained boundary delineation, achieving strong performance on segmentation benchmarks.

Despite their success, existing fusion pipelines often suffer from two fundamental limitations. First, they typically assume non-trivial *access to foundation models* (e.g., intermediate feature extraction or dense mask queries), which may be infeasible for closed-source models, API-based services, or restricted deployment scenarios. Second, their designs are highly *task- and structure-specific*: fusion mechanisms are tightly coupled with pixel-level prediction, mask generation, and spatial post-processing, and therefore do not readily extend to more general image-level tasks such as zero-shot classification, where the output is a global semantic decision rather than dense correspondence. These limitations highlight the need for a more principled, task-agnostic mechanism that makes heterogeneous foundation

---
[1]Yunnan Normal University, Kunming, China [2]Kunming University of Science and Technology, Kunming, China. Correspondence to: Zhanxuan Hu <zhanxuanhu@gmail.com>.

*Proceedings of the 43$^{rd}$ International Conference on Machine Learning*, Seoul, South Korea. PMLR 306, 2026. Copyright 2026 by the author(s).

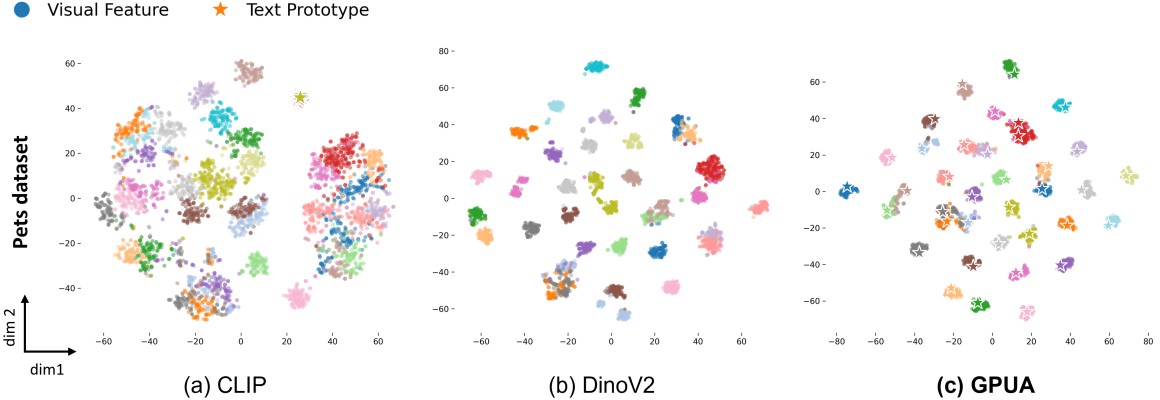

*Figure 1.* t-SNE visualization of foundation-model representations on the Pets. (a) The original CLIP space exhibits a pronounced *modality gap* between image-text embeddings. (b) VFM features yield more compact intra-class clusters, yet lack globally consistent alignment to semantic concepts. (c) **GPUA** (Ours) projects visual clusters onto their corresponding semantic anchors (⋆) while preserving intra-class structure, demonstrating effective geometry-preserving alignment and recovering accurate instance-to-prototype correspondences.

models directly compatible at the representation level.

In this work, we introduce **GPUA**, a fundamentally different framework for *unsupervised vision–language alignment*. Inspired by unsupervised cross-lingual alignment (Lample et al., 2018; Ouali et al., 2023), we view visual representations as a distinct *visual language* and recast vision–language compatibility as a cross-modal translation problem: aligning the *visual vocabulary* induced by an image encoder with the semantic vocabulary defined in a language-aligned space. Concretely, **GPUA** learns an *orthogonal transformation* that translates VFM representations into the VLM semantic space. While several prior works also attempt to bridge VFMs and VLMs by learning feature-space transformations (Ouali et al., 2023; Barsellotti et al., 2025; Jose et al., 2025), they typically rely on task-specific *supervision* and end-to-end *training*. In contrast, **GPUA** performs *fully unsupervised* alignment and learns the mapping without updating any pretrained model parameters. As illustrated in Fig. 1, the orthogonality constraint preserves the intrinsic geometry of VFM features, leading to stable alignment and effectively narrowing the modality gap.

Importantly, **GPUA** does not require a perfectly calibrated initial vision–language embedding space. Instead, it treats the language-aligned semantic space provided by a VLM as a *fixed reference* and learns a lightweight translation from any given visual space into this reference. This design makes **GPUA** naturally extensible: it can go beyond aligning a single VFM to a VLM by incorporating *multiple* visual spaces induced by heterogeneous encoders, mapping them into a shared semantic coordinate system, and performing unified inference through simple fusion. By translating and aggregating complementary perceptual geometries in the same semantic space, **GPUA** injects fine-grained visual discrimination into language-grounded recognition, effectively building a practical bridge between VLMs and VFMs.

**Contributions.** Our main contributions are three-fold: (1) We advocate *unsupervised vision–language alignment* as a principled and practical route to improve compatibility between VLMs and VFMs, and demonstrate that aligning heterogeneous foundation models is a promising direction for open-vocabulary vision. (2) We propose **GPUA**, a simple yet effective alignment framework that learns an orthogonal transformation to map VFM features into the semantic space of VLMs, without requiring labels or model parameter updates. (3) **GPUA** achieves consistent improvements across downstream benchmarks with negligible additional computation, offering a favorable accuracy–efficiency trade-off and plug-and-play deployment for zero-shot recognition and segmentation.

## 2. Related Work

### 2.1. Foundation Models

Recent years have witnessed the rise of *foundation models* trained on large-scale data, which provide general-purpose representations transferable across a wide range of tasks. In computer vision, two paradigms have become particularly influential: vision–language foundation models (VLMs) and vision-only foundation models (VFMs).

**Vision–Language Foundation Models.** VLMs such as CLIP (Radford et al., 2021) learn aligned visual and textual representations through contrastive pretraining on massive image–text corpora. A key advantage of this paradigm lies in the language-grounded semantic space, which enables training-free inference and zero-shot recognition by matching images against text prototypes. This capability has inspired research on vision–language alignment enhancement (Huang et al., 2025b), and has also been successfully extended to various downstream domains, including remote

sensing (Wang et al., 2024b; Liu et al., 2024a) and medical imaging (Lu et al., 2024). Despite their strong semantic alignment, recent studies have reported that VLM visual representations are often perceptually coarse and insufficiently sensitive to fine-grained details, particularly under distribution shift or in inductive settings.

**Vision-only Foundation Models.** In contrast, VFMs such as DINO (Oquab et al., 2023; Siméoni et al., 2025) are trained via large-scale self-supervised learning and excel at capturing intrinsic visual structures, local correspondences, and instance-level discrimination. Although VFMs exhibit strong perceptual discrimination and robust geometric structure, they still suffer from inherent limitations. Most notably, VFMs lack explicit semantic grounding and therefore cannot support open-vocabulary reasoning or text-driven inference.

### 2.2. Fusion of Foundation Models

Motivated by the complementary strengths of VFMs and VLMs, a growing body of work explores combining multiple pretrained models for improved open-vocabulary perception. Representative pipelines typically adopt CLIP as a source of language-grounded semantics, enhance it with DINO-style VFMs for fine-grained visual discrimination, and further integrate SAM-like promptable segmenters for mask-level reasoning. Such approaches have demonstrated strong empirical performance in tasks such as open-vocabulary semantic segmentation (Wysoczańska et al., 2024; Barsellotti et al., 2025; Hu et al., 2026) and vision-language grounding (Liu et al., 2024b).

### 2.3. Cross-lingual Alignment

Cross-lingual alignment is a fundamental problem in representation learning, aiming to establish compatibility between embedding spaces learned from different languages. Extensive studies in natural language processing have shown that independently learned linguistic embedding spaces can be effectively aligned by exploiting shared structural regularities, commonly instantiated through orthogonal mappings that preserve intrinsic geometric relations under the isomorphism hypothesis (Lample et al., 2018; Artetxe et al., 2018; Jawanpuria et al., 2019). These results suggest that large-scale representations typically exhibit similar geometric structures, supporting reliable alignment across domains. Inspired by this line of work, similar alignment principles have been extended beyond language to other modalities, including the alignment between visual representations and semantic concepts (Ouali et al., 2023; Schrodi et al., 2024). However, most existing approaches in the vision domain rely on labeled data, task-specific supervision, or extensive fine-tuning, which limits their applicability. *Following this principle, we further extend it to unsupervised alignment on pretrained representations.*

## 3. Method

### 3.1. Preliminaries

Our goal is to integrate the complementary strengths of Vision–Language Foundation Models (VLMs) and Vision-only Foundation Models (VFMs) in a fully *unsupervised* manner without requiring any optimization of model parameters. Specifically, we seek a lightweight mechanism that enables VFM representations, rich in perceptual geometry yet semantically ungrounded, to become directly compatible with the language-aligned semantic space of VLMs. We observe that this problem is conceptually analogous to *cross-lingual alignment* (Mikolov et al., 2013).

**Cross-lingual alignment.** In cross-lingual NLP, embeddings learned from different languages encode similar semantics but reside in heterogeneous vector spaces. Alignment is commonly achieved by learning a structure-preserving mapping that translates representations from one language into another, typically under an orthogonality constraint to maintain geometric consistency. Formally, let $\mathbf{X} \in \mathbb{R}^{N \times d}$ and $\mathbf{Y} \in \mathbb{R}^{N \times d}$ denote embeddings learned from two different languages. Cross-lingual alignment seeks a mapping $\mathbf{W} \in \mathbb{R}^{d \times d}$ by solving:

$$\min_{\mathbf{W}} \ \|\mathbf{X}\mathbf{W} - \mathbf{Y}\|_F^2, \quad \text{s.t. } \mathbf{W}^\top \mathbf{W} = \mathbf{I}, \qquad (1)$$

which admits a closed-form Procrustes solution when word-level correspondences are available.

**Unsupervised cross-lingual alignment.** A key assumption in Eq. (1) is the availability of reliable cross-lingual correspondences. In practice, however, such correspondences are often unavailable. To address this issue, unsupervised cross-lingual alignment methods (Grave et al., 2019) introduce a latent correspondence matrix $\mathbf{P}$ and jointly estimate $\mathbf{P}$ and $\mathbf{W}$:

$$\min_{\mathbf{P},\mathbf{W}} \ \|\mathbf{X}\mathbf{W} - \mathbf{P}\mathbf{Y}\|_F^2, \quad \text{s.t. } \mathbf{W}^\top \mathbf{W} = \mathbf{I}, \qquad (2)$$

where $\mathbf{P} \in \{0, 1\}^{N \times N}$ encodes *hard* alignments between the two embedding sets and each row of $\mathbf{P}$ contains exactly one non-zero entry, i.e.,

$$\sum_{j=1}^{M} P_{ij} = 1, \quad \forall i \in \{1, \ldots, N\}. \qquad (3)$$

Such formulations enable alignment without supervision by alternating between correspondence estimation and geometric mapping.

Notably, recent work (e.g., LFA (Ouali et al., 2023)) has successfully adopted Eq. (2) to mitigate the modality gap within VLMs, demonstrating strong empirical gains. However, directly transferring these unsupervised cross-lingual

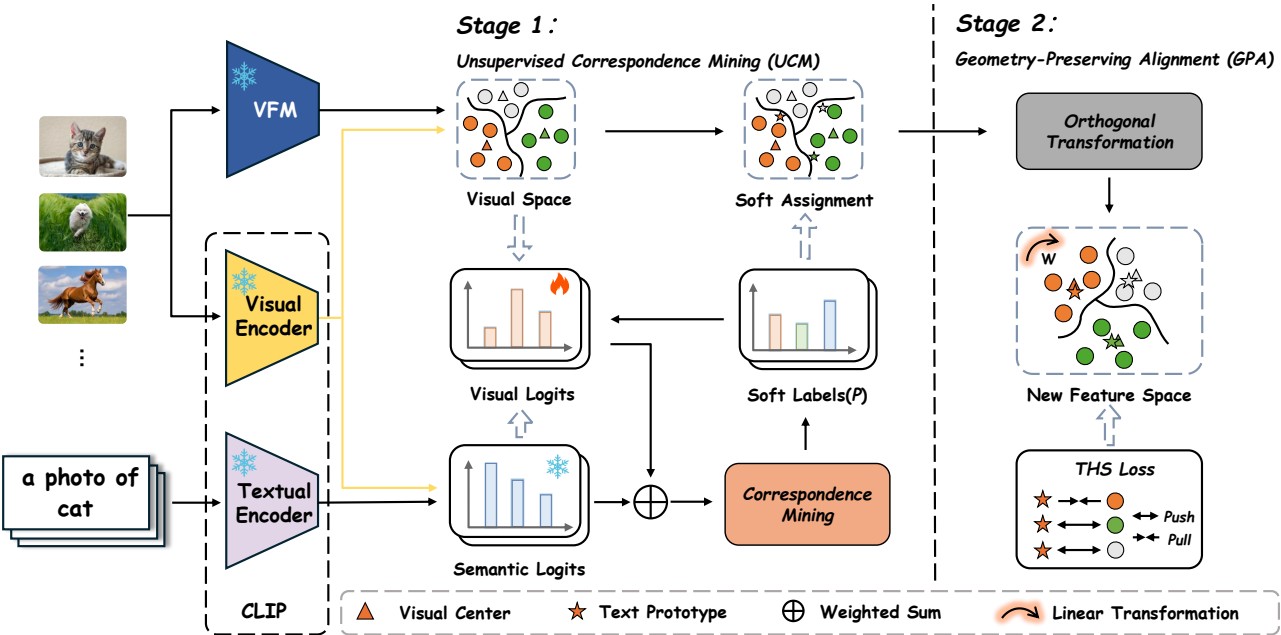

*Figure 2.* The pipeline of **GPUA**. **Stage 1**: An unsupervised correspondence estimation module infers soft assignments (**P**) by jointly enforcing structural consistency and semantic alignment between visual features and semantic prototypes. **Stage 2**: These correspondences are used to derive the optimal orthogonal transformation (**W**), which is further refined via the THS loss to yield a hubness-robust embedding space.

techniques to VFM–VLM integration is often suboptimal. First, existing formulations typically rely on alternating optimization over the mapping **W** and the correspondence matrix **P**, making the solution highly sensitive to initialization and prone to poor local optima. Second, the correspondence estimation stage often ignores the intrinsic structure of the data, which may lead to unstable or semantically inconsistent assignments under domain shift. To address these limitations, we propose a new unsupervised alignment framework, termed *Geometry-Preserving Unsupervised Alignment* (**GPUA**).

### 3.2. Geometry-Preserving Unsupervised Alignment

**Overview.** Following the cross-lingual alignment perspective, we view VFM-VLM integration as translating VFM embeddings into the language-aligned semantic space of a frozen VLM. Concretely, given VFM visual features $\mathbf{Z} \in \mathbb{R}^{N \times d_v}$ and VLM text prototypes $\mathbf{Y} \in \mathbb{R}^{K \times d_t}$, our goal is to obtain: (1) a correspondence matrix $\mathbf{P} \in \mathbb{R}^{N \times K}$ that captures (soft) instance-to-prototype associations, thereby bridging the two spaces in the absence of labels; and (2) a geometry-preserving mapping $\mathbf{W} \in \mathbb{R}^{d_v \times d_t}$ that translates VFM features into the VLM semantic space while maintaining the intrinsic geometry of VFM embeddings, enforced via an orthogonality constraint $\mathbf{W}^{\top}\mathbf{W} = \mathbf{I}$.

Unlike prior unsupervised alignment approaches that *jointly* estimate **P** and **W** through alternating optimization, we

adopt a simple *two-stage* strategy (Fig. 2). First, *Unsupervised Correspondence Mining (UCM)* infers reliable correspondences **P** by explicitly leveraging both the semantic structure provided by the VLM space and the geometric structure inherent to VFM features. Then, *Geometry-Preserving Alignment (GPA)* computes an orthogonal mapping **W** based on the mined correspondences, yielding a closed-form and stable translation from the VFM space to the VLM semantic space. *This decoupled design substantially reduces sensitivity to initialization and leads to a simple and stable alignment pipeline.*

### 3.3. Unsupervised Correspondence Mining (UCM)

We begin by revisiting *correspondence mining* in a frozen VLM semantic space. Given VLM image features $\mathbf{X} \in \mathbb{R}^{N \times d_t}$ and text prototypes $\mathbf{Y} \in \mathbb{R}^{K \times d_t}$, assigning each image to a prototype can be written as the following least-squares matching problem:

$$\min_{\mathbf{P}} \|\mathbf{X} - \mathbf{P}\mathbf{Y}\|_F^2, \quad \text{s.t. } \mathbf{P} \in \{0,1\}^{N \times K}, \ \mathbf{P}\mathbf{1} = \mathbf{1}, \quad (4)$$

where **P** is a hard assignment matrix whose $i$-th row contains exactly one non-zero entry, indicating the matched prototype for the $i$-th sample.

**Connection to $K$-means.** Interestingly, Eq. (4) is tightly connected to the standard matrix formulation of $K$-means. Let **X** denote data points, **Y** denote cluster centroids, and

$\mathbf{P}$ be the one-hot assignment matrix with $\mathbf{P1} = \mathbf{1}$. The $K$-means objective can be written as:

$$\min_{\mathbf{P,Y}} \ \|\mathbf{X} - \mathbf{PY}\|_F^2, \quad \text{s.t. } \mathbf{P} \in \{0,1\}^{N \times K}, \ \mathbf{P1} = \mathbf{1}. \quad (5)$$

Comparing Eq. (4) with Eq. (5), we observe that correspondence mining in the VLM space corresponds exactly to the *assignment step* of $K$-means when the "centroids" are fixed to be the text prototypes $\mathbf{Y}$ (see Appendix A). In other words, optimizing $\mathbf{P}$ in Eq. (4) amounts to updating cluster labels given centroids, the same operation performed in the $K$-means assignment step.

**Formulation of UCM.** The above $K$-means interpretation also clarifies a key limitation of VLM-only correspondence mining. Optimizing $\min_{\mathbf{P}} \|\mathbf{X} - \mathbf{PY}\|_F^2$ updates assignments solely by matching samples to fixed text prototypes, and thus relies heavily on semantic scores in the VLM space. However, it does not explicitly enforce that samples assigned to the same prototype form compact and geometrically coherent groups under the underlying data distribution. This issue becomes more pronounced under domain shift, where VLM similarities may be noisy and lead to unstable correspondences.

Motivated by this observation, we ask a simple question: *Can we inject the geometry-rich structure captured by VFMs into correspondence mining?* To this end, we construct a $K$-means-style structural model in the VFM space. Specifically, given VFM features $\mathbf{Z} \in \mathbb{R}^{N \times d_v}$, we introduce learnable VFM centroids $\mathbf{C} \in \mathbb{R}^{K \times d_v}$ and encourage each sample to be close to its assigned centroid. Crucially, we couple this structural view with the semantic view from the VLM by *sharing the same assignment matrix* $\mathbf{P}$ across both spaces. This naturally leads to the following unified objective:

$$\min_{\mathbf{P,C}} \ (1-\lambda)\|\mathbf{Z} - \mathbf{PC}\|_F^2 + \lambda\|\mathbf{X} - \mathbf{PY}\|_F^2,$$
$$\text{s.t.} \quad \mathbf{P} \in \{0,1\}^{N \times K}, \quad \mathbf{P1} = \mathbf{1}. \quad (6)$$

where the first term promotes *geometric coherence* in the VFM space, while the second term enforces *semantic consistency* with VLM text prototypes, and $\lambda \in [0,1]$ is trade-off parameter. By optimizing a shared $\mathbf{P}$, UCM produces correspondences that are simultaneously language-grounded and geometry-aware.

In practice, however, restricting $\mathbf{P}$ to the set of permutation (one-hot) matrices yields a combinatorial optimization problem that is intractable in practice. To obtain an efficient and stable solver, we relax $\mathbf{P}$ to a scaled assignment matrix and add an entropic regularizer. Specifically, we allow $\mathbf{P}$ to take non-negative real values and enforce its row/column marginals:

$$\mathbf{P} \in \Pi(\mathbf{r,c}) = \left\{ \mathbf{P} \in \mathbb{R}_+^{N \times K} \mid \mathbf{P1} = \mathbf{r}, \ \mathbf{P}^\top \mathbf{1} = \mathbf{c} \right\}, \quad (7)$$

where $\mathbf{r} \in \mathbb{R}_+^N$ and $\mathbf{c} \in \mathbb{R}_+^K$ specify the desired row- and column-sums. A common choice is uniform marginals, i.e., $\mathbf{r} = \frac{1}{N}\mathbf{1}$ and $\mathbf{c} = \frac{1}{K}\mathbf{1}$, which amounts to a scaled (approximately doubly-stochastic) correspondence matrix. With this relaxation, we solve the entropically regularized problem:

$$\min_{\mathbf{P} \in \Pi(\mathbf{r,c}), \, \mathbf{C}} (1-\lambda)\|\mathbf{Z} - \mathbf{PC}\|_F^2 + \lambda\|\mathbf{X} - \mathbf{PY}\|_F^2 - \varepsilon\,\mathcal{H}(\mathbf{P}), \quad (8)$$

where $\varepsilon > 0$ controls the strength of regularization and

$$\mathcal{H}(\mathbf{P}) = -\sum_{i=1}^{N}\sum_{k=1}^{K} P_{ik} \log P_{ik} \quad (9)$$

is the entropy of $\mathbf{P}$. The entropic term promotes smooth assignments and enables efficient optimization via Sinkhorn-style matrix scaling.

**Optimization of UCM.** Since the proposed objective depends on two sets of variables, namely the assignment matrix $\mathbf{P}$ and the VFM centroids $\mathbf{C}$, directly optimizing them jointly is non-trivial. We therefore adopt an alternating optimization strategy, where one variable block is updated while fixing the other, leading to a progressive reduction of the overall objective.

**Updating P.** With fixed centroids $\mathbf{C}$, optimizing Eq. (6) with respect to $\mathbf{P}$ reduces to a linear objective over the relaxed assignment space. By expanding the Frobenius norms and discarding terms independent of $\mathbf{P}$, the subproblem can be written as:

$$\max_{\mathbf{P} \in \Pi(\mathbf{r,c})} \ \langle \mathbf{P}, (1-\lambda)\mathbf{ZC}^\top + \lambda\mathbf{XY}^\top \rangle + \varepsilon\,H(\mathbf{P}), \quad (10)$$

Eq. (10) corresponds to an entropy-regularized optimal transport problem, which can be efficiently solved using the Sinkhorn–Knopp algorithm (Cuturi, 2013). The resulting correspondence matrix yields a geometry-aware soft assignment that jointly reflects structural and semantic affinities.

**Updating C.** Given the updated assignment matrix $\mathbf{P}$, the objective for the VFM centroids becomes a structural least-squares problem:

$$\min_{\mathbf{C}} \ \|\mathbf{Z} - \mathbf{PC}\|_F^2. \quad (11)$$

This formulation admits a closed-form solution where each latent centroid $\mathbf{C}_k$ is re-estimated as the weighted barycenter of the visual features:

$$\mathbf{C}_k = \frac{\sum_{i=1}^{N} P_{ik}\mathbf{Z}_i}{\sum_{i=1}^{N} P_{ik}}. \quad (12)$$

By iteratively performing these updates, UCM uncovers a coherent latent correspondence that faithfully aligns the geometric structure of visual features with the semantic topology induced by VLMs.

---

**Algorithm 1 GPUA** Optimization Algorithm

---

1: **Input:** visual features $\mathbf{Z}$, VLM visual features $\mathbf{X}$, semantic prototypes $\mathbf{Y}$, trade-off parameter $\lambda$, iterations $T$
2: **Initialization:**
3: $\mathbf{P}^{(0)} \leftarrow \text{Softmax}(\mathbf{XY}^\top); \mathbf{C}^{(0)} \leftarrow \text{update}(\mathbf{Z}, \mathbf{P}^{(0)})$
4: **Stage 1: Unsupervised Correspondence Mining**
5: **for** $t = 0$ **to** $T - 1$ **do**
6:    $\mathbf{R}^{(t)} \leftarrow (1 - \lambda)\mathbf{Z}\mathbf{C}^{(t)\top} + \lambda\mathbf{XY}^\top$
7:    $\mathbf{P}^{(t+1)} \leftarrow \text{Sinkhorn}(\mathbf{R}^{(t)})$
8:    $\mathbf{C}^{(t+1)} \leftarrow \text{update}(\mathbf{Z}, \mathbf{P}^{(t+1)})$
9: **end for**
10: # Pseudo-labels via argmax on $\mathbf{P}^{(T)}$
11: **Stage 2: Geometry-Preserving Alignment**
12: $\mathbf{W} \leftarrow \mathbf{UV}^\top$ via $\text{SVD}(\mathbf{Z}^\top \mathbf{P}^{(T)} \mathbf{Y})$
13: # Update $\mathbf{W}$ by Eq.(15)
14: $\mathbf{W} \leftarrow \mathbf{W} - \eta \nabla_\mathbf{W} \mathcal{L}_{\text{THS}}$
15: **Return** refined mapping $\mathbf{W}^*$

---

### 3.4. Geometry-Preserving Alignment (GPA)

Given the correspondence matrix $\mathbf{P}$ from Stage 1, GPA learns an orthogonal mapping $\mathbf{W}$ to translate visual features into the VLM semantic space, so that aligned features match the prototype mixture $\mathbf{PY}$:

$$\min_{\mathbf{W}} \|\mathbf{ZW} - \mathbf{PY}\|_F^2, \quad \text{s.t. } \mathbf{W}^\top \mathbf{W} = \mathbf{I}. \quad (13)$$

The orthogonality constraint enforces an (approximately) isometric translation, preventing degenerate scaling/shearing and preserving neighborhood geometry that is critical for stable nearest-prototype inference. In practice, Eq. (13) is an orthogonal Procrustes problem with a closed-form solution:

$$\mathbf{W}_0 = \mathbf{UV}^\top, \quad \mathbf{U\Sigma V}^\top = \text{SVD}\left(\mathbf{Z}^\top \mathbf{PY}\right). \quad (14)$$

Although $\mathbf{W}_0$ enables alignment without requiring model parameter updates, the aligned space may still exhibit *hubness* (Lample et al., 2018), where a few prototypes become nearest neighbors of many samples and distort the local neighborhoods. To suppress hubs, we refine $\mathbf{W}_0$ with a topology-aware ranking loss:

$$\mathcal{L}_{\text{THS}} = \frac{1}{NK} \sum_{i=1}^{N} \sum_{c \in \mathcal{N}_i^K} \left[d_i^+ + m_{i,c}^{\text{base}} + h_c - d_{i,c}\right]_+, \quad (15)$$

where $d_i^+ = \|\mathbf{W}^\top \mathbf{z}_i - \mathbf{y}_{\ell_i}\|_2$, $d_{i,c} = \|\mathbf{W}^\top \mathbf{z}_i - \mathbf{y}_c\|_2$, $\mathbf{y}_{\ell_i}$ denotes the semantic prototype corresponding to the pseudo-label $\ell_i$ of sample $i$, and $\mathcal{N}_i^K$ denotes the $K$ nearest competing prototypes. We use a semantic margin $m_{i,c}^{\text{base}} = (1 - \mathbf{y}_{\ell_i}^\top \mathbf{y}_c)/s$ and a hubness penalty

$$h_c = \frac{1}{N} \sum_{i=1}^{N} \mathbb{I}\left(c \in \mathcal{N}_i^K\right), \quad (16)$$

which up-weights margins for overly-central prototypes, discouraging them from becoming hubs. Starting from $\mathbf{W}_0$, we apply a few gradient steps on $\mathcal{L}_{\text{THS}}$ to obtain the refined mapping $\mathbf{W}^*$. The overall optimization procedure of **GPUA** is summarized in Algorithm 1.

## 4. Experiments

### 4.1. Experimental Settings

**Zero-Shot Classification.** We evaluate our method primarily on the zero-shot image classification task, which directly reflects the quality of cross-model alignment through image–text matching without task-specific training. Following the standard evaluation protocol of CLIP (Radford et al., 2021), we conduct experiments across a diverse set of public benchmarks covering different visual domains and levels of granularity. Detailed dataset descriptions and statistics are provided in Appendix C.

**Open-Vocabulary Segmentation.** We further assess the generality of our alignment framework on the open-vocabulary semantic segmentation task, which involves heterogeneous foundation models and therefore serves as a challenging testbed for cross-model compatibility. Experiments are conducted on multiple representative benchmarks, and performance is evaluated using the standard mean Intersection-over-Union (mIoU) metric. The detailed dataset information and experimental configurations are reported in Appendix C.

### 4.2. Implementation details

**General Setup.** All experiments are conducted in a fully unsupervised setting. We only require feature extraction from frozen foundation models, without updating any model parameters or introducing task-specific fine-tuning. As the visual foundation model (VFM), we adopt DINOv3 (Siméoni et al., 2025) due to its strong capability in capturing fine-grained visual structures and patch-level geometry. **GPUA** operates purely at the feature level and learns a lightweight alignment transformation on top of these frozen representations.

**Zero-Shot Classification Setting.** For zero-shot classification, **GPUA** learns a single feature transformation using the training split of each dataset, while keeping both the visual and textual encoders frozen. In this setting, we align global image representations with textual semantics, and therefore use the CLS token of the visual encoder as the image-level feature. During inference, the learned transformation is directly applied to image features, which are then matched with fixed textual prototypes encoded by the text encoder via cosine similarity. No test-time adaptation, distribution estimation, or prompt optimization is performed.

*Table 1.* **Quantitative comparison of zero-shot classification performance.** **GPUA** (Ours) leverages the full training set, while **GPUA\*** represents the performance trained with only 16 samples per class. Best results are highlighted in **bold**.

| Method | Flowers | Pets | Caltech | FGVC | EuroSAT | UCF101 | DTD | Food | Cars | SUN | ImageNet | Avg. |
|---|---|---|---|---|---|---|---|---|---|---|---|---|
| CLIP (Radford et al., 2021) | 70.7 | 89.1 | 93.2 | 24.7 | 48.3 | 67.5 | 43.5 | 85.9 | 65.6 | 62.5 | 66.6 | 65.2 |
| ZERO (Farina et al., 2024) | 67.2 | 87.8 | 94.4 | 25.2 | 42.2 | 69.2 | 45.9 | 86.8 | 69.0 | 67.6 | 71.2 | 66.0 |
| MTA (Zanella & Ben Ayed, 2024) | 68.1 | 88.2 | 94.2 | 25.2 | 45.4 | 68.7 | 45.9 | 85.0 | 68.5 | 66.7 | 70.1 | 66.0 |
| TDA (Karmanov et al., 2024) | 71.4 | 88.6 | 94.2 | 23.9 | 58.0 | 70.7 | 47.4 | 86.1 | 67.3 | 67.6 | 69.5 | 67.7 |
| ZLaP (Kalantidis et al., 2024) | 73.5 | 87.1 | 93.1 | 25.4 | 55.6 | 71.5 | 48.6 | 86.9 | 65.6 | 67.4 | 70.0 | 67.7 |
| DPE (Zhang et al., 2024a) | 75.1 | 91.1 | 94.8 | 29.0 | 55.8 | 70.4 | 54.2 | 86.2 | 67.3 | 70.1 | 71.9 | 69.6 |
| DMN (Zhang et al., 2024b) | 74.5 | 92.0 | 95.4 | 30.0 | 59.4 | 72.5 | 55.8 | 85.1 | 68.0 | 70.2 | 72.2 | 70.5 |
| StatA (Zanella et al., 2025) | 75.2 | 92.4 | 94.2 | 24.7 | 67.3 | 73.5 | 48.4 | 87.1 | 68.0 | 68.7 | 69.9 | 69.9 |
| TIPPLE (Lu et al., 2025) | 71.3 | 90.2 | 93.9 | 25.4 | 51.8 | 71.2 | 49.2 | 86.0 | 67.8 | 68.1 | 71.0 | 67.8 |
| COSMIC (Huang et al., 2025a) | 82.1 | 94.2 | 96.8 | 31.4 | 58.8 | 76.2 | 58.2 | 86.6 | 71.3 | 72.3 | **78.2** | 73.3 |
| **GPUA\*** (Ours) | **86.6** | 94.5 | **98.1** | **34.7** | 80.3 | 78.4 | 56.7 | 87.9 | 77.4 | 72.6 | 74.3 | 76.5 |
| Δ | +16.8 | +4.3 | +4.8 | +5.7 | +30.4 | +11.9 | 10.8 | +0.3 | +10.8 | +10.4 | +9.9 | +10.6 |
| **GPUA** (Ours) | 83.8 | **95.0** | 95.3 | 33.8 | **88.2** | **80.4** | **58.5** | **89.5** | **77.7** | **74.2** | 75.4 | **77.4** |
| Δ | +14.0 | +6.0 | +3.8 | +5.5 | +34.9 | +13.2 | +14.7 | +3.0 | +11.7 | +11.7 | +10.5 | +11.8 |

**Open-Vocabulary Semantic Segmentation Setting.** For open-vocabulary semantic segmentation, we evaluate **GPUA** on top of multiple representative segmentation frameworks to assess its general applicability. In contrast to image-level recognition, dense prediction requires fine-grained spatial representations. Accordingly, we perform alignment at the patch level and leverage DINOv3 patch features to enhance visual–semantic correspondence. For the vision–language models (VLMs), we consider three representative CLIP-based segmentation frameworks, namely MaskCLIP (Dong et al., 2023), SCLIP (Wang et al., 2024a), and SC-CLIP (Bai et al., 2025). These methods share a common design principle: they exploit the attention mechanisms of CLIP to improve the quality of patch-level visual features for dense prediction. This makes them a natural testbed for evaluating whether geometry-preserving alignment with a strong VFM can further enhance patch-level visual–semantic consistency.

Within each framework, **GPUA** is integrated as a plug-in alignment module by aligning DINOv3 patch-level features to the corresponding VLM semantic space. Importantly, **GPUA** does not modify the segmentation head, loss functions, or task-specific training objectives. This design isolates the effect of the proposed alignment strategy, ensuring that the observed performance gains stem from improved visual–semantic correspondence rather than architectural or optimization changes. Additional implementation details are provided in the Appendix.

### 4.3. Main Results

**Zero-shot Classification.** We evaluate **GPUA** on zero-shot image classification and compare it with representative

*Table 2.* Generalizability of **GPUA** on zero-shot semantic segmentation benchmarks.

| Method | ADE | V20 | C59 |
|---|---|---|---|
| CLIP (Radford et al., 2021) | 3.1 | 49.1 | 11.1 |
| MaskCLIP (Dong et al., 2023) | 11.9 | 54.2 | 22.2 |
| + **GPUA** (DINOv3) | 15.9 | 65.7 | 27.9 |
| SCLIP (Wang et al., 2024a) | 16.1 | 81.5 | 34.2 |
| + **GPUA** (DINOv3) | 19.1 | 87.8 | 36.3 |
| SC-CLIP (Bai et al., 2025) | 20.1 | 84.3 | 40.1 |
| + **GPUA** (DINOv3) | 21.3 | **87.6** | **41.0** |
| ProxyCLIP (Lan et al., 2024) | 19.7 | 83.0 | 37.2 |
| Talk2DINO (Barsellotti et al., 2025) | 21.1 | 87.1 | 39.8 |
| LPOSS+ (Stojnić et al., 2025) | **22.7** | 82.5 | 39.3 |

CLIP-based alignment methods across 11 benchmarks spanning diverse domains and category granularities (Table 1). Although some compared methods (e.g., the test-time adaptation approaches COSMIC (Huang et al., 2025a) and TIPPLE (Lu et al., 2025)) rely on inference-time adaptation, we include them as strong baselines under the same evaluation protocol to comprehensively assess the effectiveness of **GPUA** across different alignment paradigms. Overall, **GPUA** achieves the best performance on most datasets and improves the average accuracy by a clear margin, with notably larger gains on datasets with strong domain shift (e.g., EuroSAT) and fine-grained categories (e.g., FGVC, Cars). These results indicate that geometry-preserving alignment with a strong VFM enhances cross-model compatibility and yields more reliable visual–semantic matching. A key advantage of **GPUA** is that it learns a single orthogonal mapping *offline* from frozen model features and then applies it

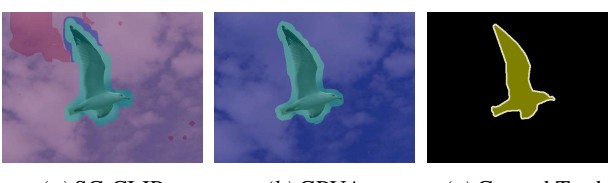

*(a) SC-CLIP*          *(b) GPUA*          *(c) Ground Truth*

*Figure 3.* Qualitative comparison on open-vocabulary semantic segmentation. (a) Predictions produced by SC-CLIP; (b) Predictions after incorporating **GPUA**; (c) Ground-truth segmentation masks. By aligning geometry-aware DINOv3 patch features with the VLM semantic space, **GPUA** enhances patch-level visual–semantic correspondence without modifying the segmentation architecture.

as a fixed translation during inference. This further suggests that the observed gains mainly stem from a higher-quality aligned embedding space itself. Finally, we report a low-data variant, **GPUA\***, where the alignment is learned from only a small set of unlabeled samples per class. Despite the limited data, **GPUA\*** remains competitive and consistently outperforms prior unsupervised alignment baselines, highlighting the robustness and sample efficiency of the proposed framework.

**Open-vocabulary Semantic Segmentation.** We further evaluate **GPUA** on open-vocabulary semantic segmentation to assess its effectiveness for dense prediction. As reported in Table 2, **GPUA** yields consistent improvements across multiple benchmarks and different CLIP-based segmentation frameworks.This is achieved without introducing any segmentation-specific architectural modifications: **GPUA** is used solely as a lightweight plug-in alignment module that leverages geometry-aware VFM (DINOv3) patch features to improve patch-level visual–semantic correspondence.

Importantly, **GPUA** does not alter the segmentation heads, loss functions, or training/inference protocols of the underlying frameworks, nor does it rely on additional task-specific modules or complex feature interaction designs adopted in methods such as (Barsellotti et al., 2025) . Therefore, the performance gains can be attributed to better aligned patch representations rather than additional task-specific engineering.Despite its simplicity, **GPUA** attains performance comparable to or exceeding methods specifically designed for open-vocabulary segmentation based on VFM representations.Qualitative comparisons are provided in Figure 3.

### 4.4. Ablation Study

We conduct an ablation study to analyze the effects of different alignment strategies in our zero-shot classification framework, with results summarized in Table 3.

**Role of VFM Choice.** We first consider a variant that removes VFM optimization, where correspondence mining

*Table 3.* Ablation study of zero-shot classification variants. ✓ indicates the activation of a component. Only-S denotes mining correspondences using semantic priors (from CLIP) only, while DINO and CLIP denote the use of their visual features.

| Only-S | CLIP | DINO | Pets | UCF101 | ImageNet |
|--------|------|------|------|--------|----------|
| ✓ | ✓ | ✓ | 89.2 | 68.3 | 71.6 |
|  | ✓ |  | 91.2 | 76.1 | 69.9 |
|  |  | ✓ | 93.1 | 75.3 | 70.6 |
|  | ✓ | ✓ | **94.5** | **78.4** | **74.3** |

*Table 4.* Ablation study of different loss functions on representative benchmarks.

| Method | EuroSAT | DTD | UCF101 | SUN | Food |
|--------|---------|-----|--------|-----|------|
| CSLS | 72.4 | 49.6 | 69.2 | 67.3 | 79.5 |
| Contrastive | 75.5 | 55.9 | 76.3 | 71.4 | 87.4 |
| Adaptive | 73.2 | 55.4 | 75.3 | 72.2 | 87.2 |
| Triplet | 75.3 | 55.4 | 76.2 | 72.5 | 87.5 |
| **THS (Ours)** | **80.3** | **56.7** | **78.4** | 72.6 | **87.9** |

is performed purely in the CLIP space by aligning CLIP visual features to CLIP text prototypes. Although this CLIP-only alignment already leads to a modest improvement, the overall gain remains limited (e.g., 89.2% on Pets, 68.3% on UCF101, and 71.6% on ImageNet). A key observation is that performance improves substantially when CLIP and DINO features are concatenated. This improvement is not merely due to feature aggregation, but rather because the combined representation exhibits higher feature quality, which in turn enables more accurate estimation of the correspondence matrix $\mathbf{P}$. More reliable correspondences directly benefit the subsequent geometry-preserving alignment, leading to a higher-quality mapping $\mathbf{W}$.

This result also highlights an important property of **GPUA**: *it naturally supports the fusion of multiple heterogeneous visual foundation models.*

**Effect of Loss Design.** We evaluate the impact of various alignment objectives in Table 4. While discriminative losses like Contrastive and Triplet improve performance over the CSLS baseline, our **THS loss** consistently yields the best results across all five representative datasets. Specifically, THS achieves a significant margin over the second-best Triplet loss on challenging tasks such as EuroSAT (**+5.0%**) and UCF101 (**+2.2%**). Notably, THS also outperforms the hubness-aware baseline Adaptive (Ouali et al., 2023), surpassing it by **7.1%** on EuroSAT and **1.3%** on DTD. These results empirically validate that explicitly suppressing hub nodes leads to a more balanced and discriminative embedding space, confirming the effectiveness of our design in robust cross-modal alignment.

# 5. Conclusion

In this paper, we propose **GPUA**, a flexible framework for unsupervised vision–language alignment that bridges vision-only and vision–language foundation models through geometry-preserving feature translation. By leveraging frozen foundation models and learning a lightweight orthogonal mapping at the feature level, **GPUA** effectively improves cross-model compatibility and delivers consistent gains on zero-shot classification and open-vocabulary semantic segmentation without test-time adaptation or task-specific tuning.

Despite its effectiveness, **GPUA** has several limitations. In particular, the correspondence estimation and alignment process does not explicitly account for data imbalance across categories, which may lead to suboptimal correspondences when class distributions are highly skewed. Addressing data imbalance and incorporating adaptive weighting or uncertainty-aware correspondence modeling constitute promising directions for future work.

## Acknowledgements

This work is supported by the Basic Research Project of Yunnan Province (Grant No. 202501CF070004), Xingdian Talent Support Program, and Intelligent Computing Center, Yunnan Normal University.

## Impact Statement

This paper presents work whose goal is to advance the field of Machine Learning. There are many potential societal consequences of our work, none which we feel must be specifically highlighted here.

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

## A. Matrix form of K-means.

Consider a dataset $\{\mathbf{x}_i\}_{i=1}^N \subset \mathbb{R}^d$ and $K$ clusters. Stack data points row-wise into

$$\mathbf{X} \triangleq \begin{bmatrix} \mathbf{x}_1^\top \\ \vdots \\ \mathbf{x}_N^\top \end{bmatrix} \in \mathbb{R}^{N \times d}.$$

Let cluster centroids be $\{\mathbf{c}_k\}_{k=1}^K \subset \mathbb{R}^d$, stacked as

$$\mathbf{C} \triangleq \begin{bmatrix} \mathbf{c}_1^\top \\ \vdots \\ \mathbf{c}_K^\top \end{bmatrix} \in \mathbb{R}^{K \times d}.$$

Define the (hard) assignment matrix $\mathbf{P} \in \{0,1\}^{N \times K}$ by

$$P_{ik} = \begin{cases} 1, & \text{if } \mathbf{x}_i \text{ is assigned to cluster } k, \\ 0, & \text{otherwise,} \end{cases} \qquad \text{s.t.} \quad \mathbf{P}\mathbf{1}_K = \mathbf{1}_N,$$

i.e., each row of $\mathbf{P}$ is one-hot.

The standard K-means objective.

$$\min_{\{\mathbf{c}_k\},\,\{\mathcal{C}_k\}} \sum_{k=1}^K \sum_{i \in \mathcal{C}_k} \|\mathbf{x}_i - \mathbf{c}_k\|_2^2 \tag{17}$$

is equivalent to the following matrix formulation:

$$\min_{\mathbf{P} \in \{0,1\}^{N \times K},\, \mathbf{C} \in \mathbb{R}^{K \times d}} \|\mathbf{X} - \mathbf{P}\mathbf{C}\|_F^2 \quad \text{s.t.} \quad \mathbf{P}\mathbf{1}_K = \mathbf{1}_N. \tag{18}$$

**Proof.** We show that the matrix formulation in (18) induces the same alternating updates as the classical K-means algorithm.

**Update of $\mathbf{P}$.** Fixing the centroids $\mathbf{C}$, the optimization over $\mathbf{P}$ becomes

$$\min_{\mathbf{P} \in \{0,1\}^{N \times K},\, \mathbf{P}\mathbf{1}=\mathbf{1}} \|\mathbf{X} - \mathbf{P}\mathbf{C}\|_F^2 = \sum_{i=1}^N \min_{k \in \{1,\ldots,K\}} \|\mathbf{x}_i - \mathbf{c}_k\|_2^2.$$

Hence, the optimal assignment is obtained by

$$P_{ik} = 1 \quad \Longleftrightarrow \quad k = \arg \min_{j \in \{1,\ldots,K\}} \|\mathbf{x}_i - \mathbf{c}_j\|_2^2,$$

which exactly coincides with the assignment step of standard K-means.

**Update of $\mathbf{C}$.** Fixing the assignments $\mathbf{P}$, the optimization over $\mathbf{C}$ becomes

$$\min_{\mathbf{C}} \|\mathbf{X} - \mathbf{P}\mathbf{C}\|_F^2.$$

This is a least-squares problem with the closed-form solution

$$\mathbf{C} = (\mathbf{P}^\top \mathbf{P})^{-1} \mathbf{P}^\top \mathbf{X},$$

or equivalently, for each cluster $k$,

$$\mathbf{c}_k = \frac{\sum_{i=1}^N P_{ik} \mathbf{x}_i}{\sum_{i=1}^N P_{ik}},$$

which is exactly the centroid update in classical K-means.

Therefore, alternating minimization of (18) with respect to $\mathbf{P}$ and $\mathbf{C}$ recovers the standard K-means algorithm. $\qquad \square$

## B. Further Analyses

**Parameter Sensitivity Analysis.**    We study the effect of the fusion coefficient $\lambda$, which controls the trade-off between geometric consistency in the VFM space and semantic alignment in the VLM space. As shown in Figure 4, classification accuracy remains relatively stable when $\lambda$ is in the range $[0.6, 0.9]$, reaching the best performance around $\lambda = 0.9$. As $\lambda$ increases, the model places greater emphasis on semantic alignment, which generally improves performance. However, when $\lambda$ becomes too large (close to 1.0), the optimization degenerates into relying almost entirely on VLM-based semantic matching, effectively weakening or removing the geometric constraints provided by VFMs. In this case, the model loses the discriminative structural information encoded in VFM features, leading to a noticeable performance drop. Similarly, when $\lambda$ is too small (e.g., $< 0.4$), the model overemphasizes geometric consistency while underutilizing semantic guidance, which also degrades performance. These results highlight the importance of jointly leveraging both geometric and semantic information rather than relying on either one alone. Based on these observations, we set $\lambda = 0.9$ as the default value for a balanced and effective alignment.

**Convergence Analysis.**    We analyze the convergence of our model by tracking classification accuracy and optimization loss over training iterations, as shown in Figure 5. Most datasets converge quickly, with accuracy stabilizing after a few iterations and loss steadily decreasing. While structured datasets such as Caltech101 and Pets reach high accuracies early, more challenging datasets like Eurosat and DTD require slightly more iterations to stabilize. Overall, the results confirm that our optimization framework converges reliably across diverse domains.

## C. Additional Implementation Details

We evaluate our method on a diverse set of benchmark datasets covering both image-level recognition and dense prediction tasks, and provide implementation details for each setting below.

**Benchmark Datasets.**    Our experiments span two tasks: zero-shot image classification and open-vocabulary semantic segmentation.

*Zero-Shot Classification.*  For image-level recognition, we evaluate on 11 widely used benchmarks with diverse visual domains and category granularities: ImageNet (Deng et al., 2009), SUN397 (Xiao et al., 2010), FGVCAircraft (Maji et al., 2013), EuroSAT (Helber et al., 2019), StanfordCars (Krause et al., 2013), Food101 (Bossard et al., 2014), Oxford-IIIT Pets (Parkhi et al., 2012), Oxford Flowers102 (Nilsback & Zisserman, 2008), Caltech101 (Fei-Fei et al., 2004), DTD (Cimpoi et al., 2014), and UCF101 (Soomro et al., 2012).

*Open-Vocabulary Segmentation.*  For dense prediction, we evaluate on three representative open-vocabulary semantic segmentation benchmarks without a background class: ADE20K (ADE) (Zhou et al., 2019), PASCAL VOC20 (Everingham et al., 2011), and PASCAL Context59 (Mottaghi et al., 2014).

**Model Configuration.**    We adopt a frozen CLIP ViT-B/16 as the base vision–language model (VLM) and DINOv3 ViT-B/16 as the vision foundation model (VFM) to provide structural priors. All model parameters are learned on an auxiliary training set and remain fixed during inference, without access to test data or any test-time optimization.

**Task-Specific Settings.**    Zero-shot classification images are center-cropped to $224 \times 224$ at inference. Class names are embedded using a single fixed prompt, e.g., "a photo of a [CLASS]", and predictions are obtained by computing the similarity between image features and class text embeddings.

Open-vocabulary segmentation training sets are constructed by extracting patch-level features using the VLM. Each semantic category leverages the semantic prior of the VLM to select the most relevant patches: specifically, the similarity between each patch feature and the corresponding text embedding is computed, and the top 1,024 patches per category with the highest similarity scores are retained to form the final training set for alignment. During inference, evaluation protocols of the corresponding baseline methods are followed, including sliding-window or multi-scale strategies when applicable.

**Experimental Details.**    During training, the orthogonal matrix $\mathbf{W}$ is obtained using a Sinkhorn-based optimal transport solver with entropy regularization $\epsilon = 0.01$. The fusion weight is set to $\lambda = 0.9$ to balance visual and semantic distributions. Visual features from VFM and VLM are combined using a simple concatenation approach. All experiments are conducted on a single NVIDIA RTX 4090 GPU.

*Table 5.* **Effect of different VFM and VLM combinations on zero-shot classification using GPUA.** The upper block reports results under the 16-shot per-class training setting, while the lower block uses the full training set.

| Method | Flowers | Pets | Caltech | FGVC | EuroSAT | UCF101 | DTD | Food | Cars | SUN | ImageNet | Avg. |
|---|---|---|---|---|---|---|---|---|---|---|---|---|
| *low-data setting (16 samples per class)* | | | | | | | | | | | | |
| Only DINOv2 | 86.52 | 91.61 | 97.65 | 28.32 | 67.49 | 77.24 | 53.01 | 81.51 | 71.67 | 68.05 | 73.23 | 72.39 |
| DINOv2 + CLIP | 87.5 | 93.4 | 98.0 | 30.4 | 78.7 | 79.4 | 54.3 | 86.2 | 76.4 | 72.9 | 76.5 | 75.8 |
| Only DINOv3 | 86.6 | 93.1 | 97.3 | 33.7 | 83.9 | 75.3 | 55.9 | 85.5 | 76.4 | 67.4 | 70.6 | 75.1 |
| DINOv3 + CLIP | 84.1 | 94.5 | 98.0 | 33.4 | 75.7 | 78.4 | 55.4 | 88.0 | 77.4 | 72.2 | 74.3 | 75.6 |
| *Full training set* | | | | | | | | | | | | |
| DINOv2 + CLIP | 84.7 | 94.7 | 97.0 | 30.5 | 83.2 | 80.9 | 58.2 | 88.5 | 77.3 | 74.2 | 77.1 | 76.9 |
| DINOv3 + CLIP | **83.8** | **95.0** | **95.3** | **33.8** | **88.2** | **80.4** | **58.5** | **89.5** | **77.7** | **74.2** | **75.4** | **77.4** |

*Table 6.* Comparison of different alignment losses on zero-shot classification benchmarks.

| Method | Flowers | Pets | Caltech | Aircraft | EuroSAT | UCF101 | DTD | Food | Cars | SUN | ImageNet | Avg. |
|---|---|---|---|---|---|---|---|---|---|---|---|---|
| CSLS | 83.1 | 83.1 | 92.2 | 31.5 | 72.4 | 69.2 | 49.6 | 79.5 | 69.7 | 67.3 | 66.7 | 69.5 |
| Adaptive | 85.1 | 94.0 | 98.1 | 34.0 | 73.2 | 75.3 | 55.4 | 87.2 | 76.6 | 72.2 | 73.9 | 75.0 |
| Contrastive | 85.6 | 94.4 | 97.9 | 33.3 | 75.5 | 76.3 | 55.9 | 87.4 | 76.2 | 71.4 | 72.7 | 75.1 |
| Triplet | 85.3 | 93.7 | 98.0 | 33.7 | 75.3 | 76.2 | 55.4 | 87.5 | 77.0 | 72.5 | 74.1 | 75.3 |
| THS | 86.6 | 94.5 | 98.1 | 34.7 | 80.3 | 78.4 | 56.7 | 87.9 | 77.4 | 72.6 | 74.3 | 76.5 |

# D. Additional GPUA Results

In this section, we present additional experiments, as summarized in Tables 6 to 8 and Figures 4 to 5.

**Effect of backbone architectures.** The COSMIC baseline reported in Table 1 adopts a DINOv2-based visual backbone following its original implementation. To ensure a fair comparison, we further evaluate the impact of different visual foundation model (VFM) and vision–language model (VLM) combinations in Table 5. In particular, when using the same backbone architecture as COSMIC (e.g., DINOv2), **GPUA** still consistently outperforms the baseline across datasets and training settings. These results indicate that the performance improvements of **GPUA** mainly originate from the proposed alignment strategy rather than simply benefiting from stronger backbone architectures such as DINOv3. Moreover, **GPUA** demonstrates stable performance across different backbone combinations, suggesting that the proposed method is largely model-agnostic and can generalize effectively across heterogeneous representation spaces.

**Generalizability across domains.** We further evaluate **GPUA** on multiple ImageNet-related benchmarks, including ImageNet, ImageNet-A, ImageNet-V2, ImageNet-R, and ImageNet-S. As shown in Table 7, **GPUA** consistently improves performance across all benchmarks, increasing the average accuracy from 59.1 to 67.3 compared with CLIP. In particular, **GPUA** achieves notable gains on challenging distribution-shift benchmarks such as ImageNet-A and ImageNet-S, demonstrating strong cross-dataset generalization. The generalization capability of **GPUA** mainly benefits from the strong visual–semantic representations provided by large-scale pretrained foundation models. Building upon these representations, **GPUA** further enhances cross-modal feature consistency while preserving the original semantic structure, leading to more reliable visual–semantic matching across different data distributions.

**Extension to Dense Prediction Tasks.** We further evaluate **GPUA** on additional zero-shot semantic segmentation benchmarks to assess its generalizability on dense prediction tasks. As shown in Table 8, we extend the evaluation to COCO-Stuff164K (Caesar et al., 2018) and Cityscapes (Cordts et al., 2016), in addition to ADE20K, V20, and C59. **GPUA** consistently achieves the best performance across all datasets, outperforming both SC-CLIP and Talk2DINO. Notably, simply incorporating stronger visual foundation model (VFM) features does not necessarily guarantee improved segmentation performance. For example, the prior VFM-based method Talk2DINO performs comparably to or even worse than SC-CLIP on several datasets. In contrast, **GPUA** consistently improves performance over both methods across all benchmarks, including further gains on Stuff164K and Cityscapes. These results suggest that the improvements mainly originate from more effective cross-modal feature alignment rather than solely from stronger backbone representations.

*Table 7.* Out-of-Domain Generalization: Comparison between CLIP and **GPUA** on ImageNet Benchmarks.

| Method | ImageNet | ImageNet-A | ImageNet-V2 | ImageNet-R | ImageNet-S | Average |
|--------|----------|------------|-------------|------------|------------|---------|
| CLIP | 66.7 | 47.9 | 60.9 | 74.0 | 46.1 | 59.1 |
| **GPUA** | **76.5** | **57.4** | **68.2** | **77.5** | **56.9** | **67.3** |

*Table 8.* Extension of **GPUA** to Zero-Shot Semantic Segmentation Benchmarks.

| Method | ADE20K | V20 | C59 | Cityscapes | Stuff164K |
|--------|--------|-----|-----|------------|-----------|
| SC-CLIP | 20.1 | 84.3 | 40.1 | 41.0 | 26.9 |
| **GPUA** | **21.3** | **87.6** | **41.0** | **42.0** | **29.0** |
| Talk2DINO | 21.1 | 87.1 | 39.8 | 36.6 | 28.1 |

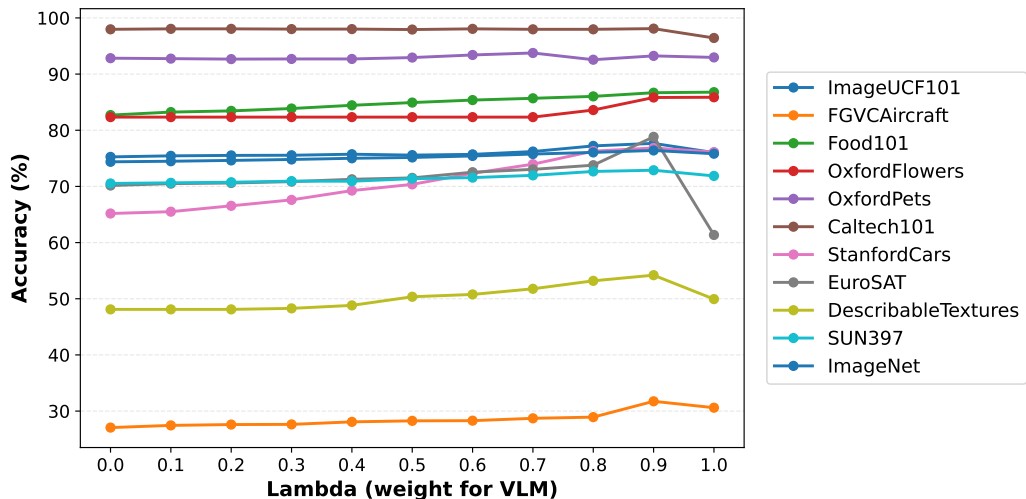

*Figure 4.* **Sensitivity analysis of the fusion coefficient $\lambda$.** Classification accuracy (%) versus $\lambda$ across 11 datasets, showing optimal performance around $\lambda = 0.9$.

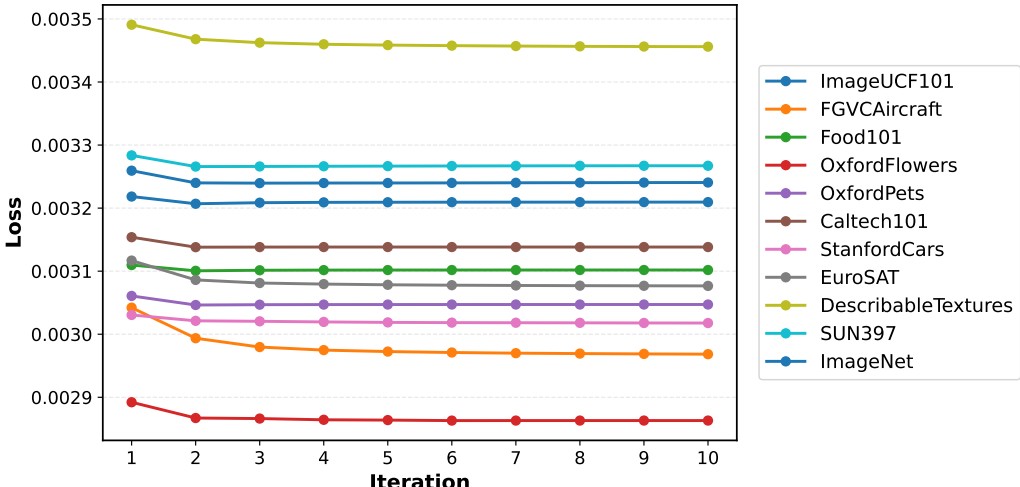

*Figure 5.* Convergence of the proposed method on 11 datasets. The method converges within a few iterations, indicating efficient and stable optimization.

