# OpenReview forum: "Geometry-Preserving Unsupervised Alignment for Heterogeneous Foundation Models"
_ICML.cc/2026/Conference — ICML 2026 regular_

### Official Review · Reviewer_YgqF · 2026-03-11

**Soundness:** 3
**Presentation:** 3
**Significance:** 3
**Originality:** 3
**Overall Recommendation:** 5
**Confidence:** 4

**Summary:**

This paper introduces GPUA, a two-stage method for aligning vision foundation models with vision–language models. The proposed approach removes the need for complex intermediate feature extraction, making it suitable for closed-source foundation models. In addition, unlike prior methods, GPUA adopts a task-agnostic design that enables broader applicability across tasks. The authors evaluate the method on zero-shot image classification and open-vocabulary segmentation, showing consistent performance improvements over standard CLIP and several test-time adaptation baselines.

**Compliance With Llm Reviewing Policy:**

Affirmed.

**Final Justification:**

The authors provided a strong and clear rebuttal that resolved my questions.

**Key Questions For Authors:**

Questions for Authors are derived from the weaknesses:
1. To substantiate the "training-free" claim, how well does the mapping matrix $W$ generalize across datasets? Can the authors provide a cross-dataset experiment where $W$ is computed on a source distribution (e.g., ImageNet) and evaluated on target datasets (e.g., EuroSAT or FGVC) without recomputation?

2. Given their methodological similarities, why is Talk2DINO excluded from the classification benchmarks? Can the authors provide the missing evaluations to ensure a fair baseline assessment?

3. What specific visual backbone is utilized by the COSMIC baseline in Table 1? Clarification is needed here to determine whether GPUA's gains stem from the alignment method itself or simply the use of the DINOv3 backbone.

**Limitations:**

yes

**Strengths And Weaknesses:**

### Strengths

1. This paper proposes a task-agnostic and practical alignment method that works even with closed-source vision foundation models by eliminating the need for complex intermediate feature extraction, thereby overcoming key limitations of prior methods.

2. The method shows solid empirical improvements in zero-shot classification, particularly on fine-grained and domain-shifted datasets. The low-data variant demonstrates reasonable sample efficiency.

3. The paper is clear and well-written, the proposed method is correctly formalized and the figures go straight to the point and help the understanding.

### Weaknesses

1. The core framing of the method as "training-free" is misleading. Since GPUA requires a dataset-specific mapping $W$ computed over the target distribution, it is fundamentally a transductive approach rather than a universal alignment. Also, the lack of cross-dataset generalization experiments leaves the transferability of the learned mapping unclear.

2. The baseline comparisons are incomplete. Talk2DINO [1] learns a similar mapping from DINO to CLIP space, yet it is excluded from the classification benchmarks while demonstrating competitive performance on the segmentation datasets provided. Furthermore, the COSMIC [2] baseline in Table 1 is ambiguous; without knowing whether COSMIC also leverages DINOv3, it is difficult to determine whether GPUA's gains stem from the alignment strategy itself or simply from the use of a stronger visual backbone.

3. The segmentation evaluation is too narrow to firmly establish the method's efficacy for dense prediction tasks. Testing on only three benchmarks and plugging into only three CLIP variants limits the scope of the claims. The performance gains over SC-CLIP are marginal, and the omission of standard open-vocabulary segmentation benchmarks like COCO-Stuff and Cityscapes weakens the evaluation.

\
[1] Talking to DINO: Bridging Self-Supervised Vision Backbones with Language for Open-Vocabulary Segmentation (Barsellotti et al., 2025)

[2] COSMIC: Clique-Oriented Semantic Multi-space Integration for Robust CLIP Test-Time Adaptation (Huang et al., 2025)

---

> ### Author Rebuttal · Authors · 2026-03-31
>
> We sincerely thank the reviewer for the insightful and constructive comments, which have helped us improve our work. Due to space limitations, some parts of the responses have been omitted.
>
> >Q1. To substantiate the ``training-free'' claim, ...
>
> A1. Regarding generalization, we agree that both CLIP and our method may exhibit limited transferability when applied to unseen domains without access to target data. Importantly, the strong generalization ability of large VLMs (e.g., CLIP) stems from training on massive and diverse datasets. However, it is also well known that CLIP's performance degrades under domain shift or on fine-grained categories.
> Similarly, our alignment is learned from limited data, and its generalization is naturally bounded by the diversity of the data used. Nevertheless, when the target categories are relatively aligned with the source domain, our method still demonstrates strong generalization ability. As shown in Table 1, when learning the mapping on ImageNet and evaluating on related datasets (e.g., ImageNet-S), GPUA achieves consistent improvements, indicating that the learned alignment can transfer across semantically similar domains.
> We will clarify this point more explicitly in the revision.
>
> Table 1. Comparison on ImageNet and OOD datasets
> | Method | ImageNet | ImageNet-A | ImageNet-V2 | ImageNet-R | ImageNet-S | Average |
> | ----- | :------: | :--------: | :---------: | :--------: | :--------: | :-----: |
> | CLIP  |   66.7   |    47.9    |     60.9    |    74.0    |    46.1    |   59.1  |
> | GPUA  |   76.5   |    57.4    |     68.2    |    77.5    |    56.9    |   67.3  |
>
> >Q2. Given their methodological similarities, ...
>
> A2. We thank the reviewer for this suggestion. While Talk2DINO shares a similar motivation of bridging VFM and VLM, it is primarily designed for dense prediction tasks (e.g., open-vocabulary segmentation) and operates on patch-level representations. In contrast, our classification setting relies on global image-level features, leading to fundamentally different input–output formats and evaluation protocols. Therefore, Talk2DINO is not directly applicable to standard classification benchmarks without substantial modification.
>
> >Q3. What specific visual backbone is utilized by the COSMIC baseline in Table 1? Clarification is needed here to determine whether GPUA's gains stem from the alignment method itself or simply the use of the DINOv3 backbone.
>
> A3. COSMIC in Table 1 uses a DINOv2-based visual backbone (consistent with its original implementation).
> To ensure a fair comparison, we further evaluate the impact of different backbones in Table 6. In particular, when using the same backbone as COSMIC (e.g., DINOv2), our method still consistently outperforms COSMIC.
> This demonstrates that the performance gains of GPUA mainly come from the proposed alignment strategy, rather than the choice of a stronger backbone (e.g., DINOv3). We will clarify this point more explicitly in the revision.
>
> >Q4. The segmentation evaluation is too narrow to firmly establish the method's efficacy for dense prediction tasks...
>
> A4. We thank the reviewer for the comments on the evaluation scope and the effectiveness of our method on dense prediction tasks. We agree that evaluating on more benchmarks would further strengthen the generality of our conclusions. Accordingly, we additionally include results on COCO-Stuff164K and Cityscapes (see Table 2) to extend the evaluation.
> Regarding the use of more CLIP variants, GPUA is a model-agnostic feature alignment method that does not depend on specific architectural designs, but primarily relies on the quality of the extracted features. Therefore, we do not further extend the evaluation in this direction.
> Regarding the concern about limited gains over SC-CLIP, simply incorporating VFM features does not guarantee better segmentation performance. As shown in Table 2, prior VFM-based methods (e.g., Talk2DINO) perform comparably to or worse than SC-CLIP on some datasets, indicating that improvements cannot be explained solely by stronger visual backbones. In contrast, GPUA operates purely at the feature alignment level and consistently improves performance across datasets, indicating that the gains stem from more effective cross-modal alignment.
> We will clarify these points in the revised version.
>
> Table 2. Generalizability of GPUA on zero-shot semantic segmentation benchmarks
> | Method    | ADE20K    | V20       | C59       | Cityscapes | Stuff164K |
> | :-------: | :-------: | :-------: | :-------: | :--------: | :-------: |
> | SC-CLIP   | 20.1      | 84.3      | 40.1      | 41.0       | 26.9      |
> | GPUA      | **21.3** | **87.6** | **41.0** | **42.0**   | **29.0**  |
> | Talk2DINO | 21.1      | 87.1      | 39.8      | 36.6       | 28.1      |

---

> > ### Author Rebuttal · Reviewer_YgqF · 2026-04-04
> >
> > Thank you for your rebuttal and for addressing my concerns. I'd like to maintain my score.

---

> > > ### Author Response · Authors · 2026-04-04
> > >
> > > Thank you for your feedback. We will incorporate the relevant additions into the revised manuscript.

---

### Official Review · Reviewer_iy8s · 2026-03-12

**Soundness:** 4
**Presentation:** 4
**Significance:** 3
**Originality:** 4
**Overall Recommendation:** 5
**Confidence:** 5

**Summary:**

This paper studies visual recognition from the perspective of treating vision as a language, and proposes a corresponding framework for visual representation and reasoning. The approach is validated through extensive experiments and ablation studies.

**Compliance With Llm Reviewing Policy:**

Affirmed.

**Final Justification:**

My concerns have been addressed by the authors’ response.

**Key Questions For Authors:**

1. Why are P and W not learned jointly?.
2. What is the relationship between Eq. (4) and Eq. (5)? Also, why is Y not updated in Eq. (6)?
3. In the parameter analysis, the performance increases monotonically with \lambda but drops when \lambda = 1. The paper does not provide a clear explanation for this phenomenon？
4. Why is the constraint in Eq. (7) introduced?
5. Figure 2 is not very clear and could be improved.

**Limitations:**

Yes.

**Strengths And Weaknesses:**

Strengths:
1. The paper is well written.
2. The concept of treating vision as a language is intriguing and provides an interesting perspective for VFM and VLM.
3. The proposed method is well supported by extensive experiments and ablation studies.

Weaknesses:
1. The paper lacks analysis of convergence behavior and computational complexity.
2. The process of learning W is a linear transformation.
3. Some recent works on vision–language models are missing in the related work or experimental comparison.

---

> ### Author Rebuttal · Authors · 2026-03-30
>
> >Q1. The paper lacks analysis of convergence behavior and computational complexity.
>
> A1.We thank the reviewer for this valuable suggestion. Regarding convergence, we provide an empirical analysis in the supplementary material (Fig. 4), where the optimization of $P$ typically converges within fewer than 10 iterations, demonstrating stable and efficient behavior in practice.
> In terms of computational complexity, the main cost comes from the Sinkhorn iterations for $P$ and matrix operations for computing $W$. We will include a more detailed analysis in the revision.
>
> >Q2. The process of learning W is a linear transformation.
>
> A2. Please see the A2 of "Response to Reviewer MXfM".
>
> >Q3. Some recent works on vision–language models are missing in the related work or experimental comparison.
>
> A3.  Thank you for the question. We will further update the related work and include additional relevant methods to provide a more comprehensive comparison in the revision.
>
> >Q4. Why are P and W not learned jointly?
>
> A4. While jointly optimizing $P$ and $W$ is a natural formulation (as in prior cross-lingual alignment works), it typically leads to a highly non-convex problem and is sensitive to initialization, often resulting in unstable or suboptimal solutions. Instead, we adopt a decoupled strategy: we first estimate a reliable correspondence matrix $P$ via geometry-aware matching, and then compute $W$ in closed form (orthogonal Procrustes). This design significantly improves stability and avoids error amplification from noisy initialization. Empirically, we observe that this two-stage approach leads to more robust alignment compared to joint optimization.
>
> >Q5. What is the relationship between Eq. (4) and Eq. (5)? Also, why is Y not updated in Eq. (6)?
>
> A5. Eq. (4) can be viewed as a special case of the K-means formulation in Eq. (5), where the cluster centroids are fixed as the text prototypes $Y$. In this sense, optimizing Eq. (4) corresponds to the assignment step of K-means, assigning each sample to its closest semantic prototype.
> In our formulation, $Y$ represents semantic prototypes derived from the VLM and is treated as a fixed reference space. Updating $Y$ would alter the semantic structure and break the connection to the language-aligned space provided by the VLM. Instead, we introduce learnable visual centroids $C$ in the VFM space and optimize them jointly with the correspondence matrix $P$, while keeping $Y$ fixed to preserve semantic consistency.
> We will clarify these points more explicitly in the revision.
>
> >Q6. In the parameter analysis, the performance increases monotonically with $\lambda$, but drops when $\lambda = 1$. The paper does not provide a clear explanation for this phenomenon.
>
> A6.  The parameter $\lambda$ controls the trade-off between geometric consistency (VFM space) and semantic alignment (VLM space). As $\lambda$ increases, the model places more emphasis on semantic alignment, which generally improves performance. However, when $\lambda = 1$, the objective reduces to relying purely on VLM-based semantic matching (i.e., the geometric constraint from VFM is completely removed). In this case, the model loses the discriminative structure provided by VFM features, leading to a performance drop. This behavior highlights the importance of jointly leveraging both geometric and semantic information, rather than relying on either alone. We will clarify this phenomenon more explicitly in the revision.
>
> >Q7. Why is the constraint in Eq. (7) introduced?
>
> A7. Thank you for the question. The constraint in Eq.~(7) is mainly introduced to enable efficient optimization of $P$ using Sinkhorn iterations. Directly enforcing $P$ to be a one-hot matrix leads to a combinatorial optimization problem that is intractable. Therefore, we relax $P$ to a non-negative matrix with prescribed row and column marginals, i.e., $P \in \Pi(r, c)$. This relaxation not only allows continuous optimization but also naturally fits the Sinkhorn algorithm, yielding smooth and stable assignment results.
>
> >Q8. Figure 2 is not very clear and could be improved.
>
> A8. In the revision, we will refine the figure by simplifying the pipeline illustration, enlarging key components, and providing more explicit annotations to better convey the overall framework and data flow.

---

> > ### Author Rebuttal · Reviewer_iy8s · 2026-04-01
> >
> > The  response of authors was very comprehensive and addressed my previous concerns, I'm willing to improve my score.

---

> > > ### Author Response · Authors · 2026-04-03
> > >
> > > Thank you for your positive feedback and for improving your score. We will incorporate the relevant clarifications into the revised manuscript.

---

### Official Review · Reviewer_MXfM · 2026-03-12

**Soundness:** 2
**Presentation:** 2
**Significance:** 2
**Originality:** 2
**Overall Recommendation:** 4
**Confidence:** 4

**Summary:**

This paper proposes a geometry-preserving unsupervised alignment framework for integrating heterogeneous foundation models named GPUA. Specifically, GPUA consists of two core components: the Unsupervised Correspondence Mining module (UCM) and the Geometry-Preserving Alignment module (GPA). The UCM module captures reliable soft correspondences between visual features and semantic prototypes through entropy-regularized optimal transport and joint optimization of VLM semantic structure and VFM geometric structure. The GPA module enhances cross-modal compatibility via a closed-form orthogonal Procrustes solution that translates VFM features into the VLM semantic space while preserving intrinsic neighborhood geometry.

The experimental evaluation in this paper assesses the effectiveness of GPUA across both classification and open-vocabulary segmentation benchmarks. Results demonstrate that GPUA achieves zero-shot classification gains while maintaining training-free and parameter-efficient deployment, outperforming several mainstream CLIP adaptation and cross-modal alignment approaches.

**Compliance With Llm Reviewing Policy:**

Affirmed.

**Final Justification:**

The authors have well addressed my concerns and all reviewers have recognized the contributions of this paper and the good results. So, I decide to improve my rating to 4.

**Key Questions For Authors:**

Please refer to the weaknesses part.

**Limitations:**

The authors discuss the limitations of this paper.

**Strengths And Weaknesses:**

Strengths
- Attempting to integrate the complementary strengths of VFMs and VLMs through a training-free paradigm is considered novel.
- Performing lightweight alignment at the feature level does not impact downstream pipelines.
- GPUA achieves consistent improvements on zero-shot classification benchmarks.

Weaknesses
- The correspondence matrix P is based on the assumption that the VFM geometric structure and the VLM semantic structure have the same clustering partition. However, when the domain shift is large or the category granularity is inconsistent, this assumption may fail, leading to incorrect correspondence. Although entropy regularization and Sinkhorn are applied to improve robustness, they still rely on heuristic soft matching.
- This paper learns an orthogonal transformation matrix W via an objective function to transform the VFM feature space into the VLM semantic space. Although the training-free orthogonal matrix can preserve geometric structure, it also limits the flexibility of the mapping. When there are non-linear correlations between VFM and VLM spaces, the linear orthogonal transformation may not align sufficiently.
- Although GPUA is training-free, it still requires the target dataset's training set images to learn the mapping. Appendix Table 6 shows that using the matrix W learned from ImageNet directly on other datasets, the performance is significantly lower than "learning separately per dataset". I understand the author's claim that this is still higher than the original CLIP, but this reliance on target-specific alignment limits its generalizability in strict zero-shot classification scenarios where target images are inaccessible.
- The Sinkhorn algorithm assumes class uniform distribution; therefore, when facing non-uniform distributions (even long-tail distributions), forced uniform assignment may lead to minority class samples being incorrectly assigned to majority class prototypes, thereby affecting the linear mapping.

---

> ### Author Rebuttal · Authors · 2026-03-31
>
> We sincerely thank the reviewer for the insightful and constructive comments, which have helped us improve our work. Due to space limitations, some parts of the responses have been omitted.
>
> >Q1. The correspondence matrix P is based on the assumption that the VFM geometric structure and the VLM semantic structure have the same clustering partition, ...
>
> A1. We thank the reviewer for this insightful comment. We agree that under severe domain shift, if the underlying VFM or VLM themselves fail to generalize, a training-free method like ours may also degrade, since we do not update model parameters.
> However, this limitation is inherent to all methods that rely on frozen foundation models. In practice, many domains (e.g., medical imaging, remote sensing) already have specialized VFMs or VLMs with reasonable domain coverage. Our framework is orthogonal to model choice and can be directly applied to such domain-specific models to further improve cross-modal alignment.
>
> >Q2. This paper learns an orthogonal transformation matrix W via an objective function to transform the VFM feature space into the VLM semantic space. Although the training-free orthogonal matrix can preserve geometric structure, it also limits the flexibility of the mapping,...
>
> A2. We agree that a linear orthogonal mapping is less expressive than non-linear transformations.
> However, learning a non-linear mapping would still require establishing reliable correspondences between visual features and semantic prototypes. In fact, \emph{how to determine such cross-modal dependencies without supervision} is a core challenge, and precisely one of the main contributions of our work.
> Our design, therefore, decouples this problem: we first estimate robust correspondences in an unsupervised manner, and then adopt a simple orthogonal mapping to ensure stable and geometry-preserving alignment. This avoids introducing additional parameters or overfitting under the training-free setting. Empirically, we show that this combination is sufficient to achieve strong performance across diverse datasets.
>
> >Q3. Although GPUA is training-free, it still requires the target dataset's training set images to learn the mapping. Appendix Table 6 shows that using the matrix W learned from ImageNet directly on other datasets, the performance is significantly lower than "learning separately per dataset",...
>
> A3. First, we would like to clarify that Table 6 reports the effect of different VFM–VLM combinations under both low-data and full training settings, rather than cross-dataset transfer as suggested.
> Regarding generalization, we agree that both CLIP and our method may exhibit limited transferability when applied to unseen domains without access to target data. Importantly, the strong generalization ability of large VLMs (e.g., CLIP) stems from training on massive and diverse datasets. However, it is also well known that CLIP's performance degrades under domain shift or on fine-grained categories.
> Similarly, our alignment is learned from limited data, and its generalization is naturally bounded by the diversity of the data used. Nevertheless, when the target categories are relatively aligned with the source domain, our method still demonstrates strong generalization ability. As shown in Table 1, when learning the mapping on ImageNet and evaluating on related datasets (e.g., ImageNet-S), GPUA achieves consistent improvements, indicating that the learned alignment can transfer across semantically similar domains.
>
> Table 1. Comparison on ImageNet and OOD datasets
> | Method | ImageNet | ImageNet-A | ImageNet-V2 | ImageNet-R | ImageNet-S | Average |
> | ----- | :------: | :--------: | :---------: | :--------: | :--------: | :-----: |
> | CLIP  |   66.7   |    47.9    |     60.9    |    74.0    |    46.1    |   59.1  |
> | GPUA  |   76.5   |    57.4    |     68.2    |    77.5    |    56.9    |   67.3  |
>
>
> >Q4. The Sinkhorn algorithm assumes class uniform distribution; therefore, when facing non-uniform distributions (even long-tail distributions), ...
>
> A4. We agree that the standard Sinkhorn formulation with uniform marginals may be suboptimal under non-uniform or long-tail distributions. To address this, our framework can be naturally extended to \emph{unbalanced optimal transport}, which relaxes the marginal constraints and allows more flexible matching. In addition, in practical scenarios with sufficient data, simple strategies such as class-balanced sampling can effectively mitigate this issue. Empirically, as shown in Table 6, even under a low-data setting with only 16 unlabeled samples per class, our method is able to learn a strong alignment function, suggesting that the proposed approach is robust to moderate data imbalance.

---

> > ### Author Rebuttal · Reviewer_MXfM · 2026-04-01
> >
> > The authors have solved my concerns. The authors are encouraged to include the new reponses in the revision. I would like to raise my score rating.

---

> > > ### Author Response · Authors · 2026-04-03
> > >
> > > Thank you for your feedback and for raising your score. We will incorporate the new responses into the revised manuscript to further improve the quality of the paper.

---

### Official Review · Reviewer_G3hR · 2026-03-13

**Soundness:** 1
**Presentation:** 1
**Significance:** 1
**Originality:** 2
**Overall Recommendation:** 4
**Confidence:** 4

**Summary:**

This work aims marriage vision-language models (VLM) and vision-only foundation models (VFM) for keeping their respective advantages. The idea is to translate the VFM space to the VLM space via an orthogonal mapping, with the hope of preserving geometry while bridging the modality gap. This method just needs to learn this mapping, agnostic to the models and downstream tasks.

**Compliance With Llm Reviewing Policy:**

Affirmed.

**Final Justification:**

The authors have addressed well my concerns about the setting, main idea, definition, presentation etc. These issues are critical to be addressed in the final version, to have better readability and ease exposure to the general audience. This work adds value to the unsupervised alignment of foundation models for task adaption, after reading the rebuttal.

**Key Questions For Authors:**

Please see above the weaknesses.

**Limitations:**

Yes

**Strengths And Weaknesses:**

**Weaknesses**

- Abstract cannot serve the purpose of giving the picture due to these issues:
> - The position of this whole work is misleading: from the starting sentences and the title, it seems to suggest this work to build a new foundation model. However, this is about how to combine multiple foundation models (VLM and VFM) for some downstream tasks.
> - What is the connection between preserving geometry and combining the advantages of VFM and VLM?
> - It is unclear why an orthogonal mapping between the two space could keep the geometry and also what geometry is kept here;
> - Here what data is used for learning this mapping is unclear at all.

- Introduction:
>- When discussing previous works in the 2nd paragraph, it is suggested that the specific reference is made so that readers can understand better.

- Novelty limitation: Leaning a mapping from vision space to semantic/text space has been studied extensively, with one of the early works is Flamingo [R1] and follow up works. I do not see any novelty idea with this at this moment. Indeed, there are more options for VFM and VLM models but that does not form good novelty even giving better performances.

- Method:
> - While it is claimed to be unsupervised, this method needs to provide a set of text prompts which are assumed to be available. That means it is not fully unsupervised
> - What means by geometry structure in this context? It is hard to follow on page 5.
> - Overall, this method section is hard to understand except telling that what is involved inside this method.

- Evaluation:
>- It is rather confusing that some test time adaptation methods (e.g., COSMIC, TIPPLE) is compared in Table 1. Is this work doing the same setting? It seems this work is messed up from problem, writing, all way to the evaluation setting.


**References:**
[R1] Flamingo: a Visual Language Model for Few-Shot Learning. NeurIPS 2022

---

> ### Author Rebuttal · Authors · 2026-03-31
>
> We sincerely thank the reviewer for the insightful and constructive comments, which have helped us improve our work.
> Due to space limitations, some parts of the responses have been omitted.
>
> >Q1. The position of this whole work is misleading，...
>
> A1. We thank the reviewer for the comment. Our work does not aim to build a new foundation model, but focuses on unsupervised alignment between the visual and language spaces of existing foundation models.
> Both the title (“unsupervised alignment”) and the abstract emphasize a feature-level transformation (orthogonal mapping) without updating model parameters. Therefore, GPUA is a compatibility framework, rather than a new foundation model.
>
> >Q2. What is the connection between preserving geometry and combining the advantages of VFM and VLM?
>
> A2. As stated in the abstract, VFMs provide discriminative perceptual geometry, while VLMs provide language-grounded semantic alignment. Our goal is to combine these complementary properties.
> Specifically, the geometry-preserving constraint ensures that when VFM features are mapped into the VLM semantic space, their intrinsic structural relationships (e.g., neighborhood and cluster structure) are retained. This allows VFM’s fine-grained discriminative geometry to be preserved while aligning it with VLM’s semantic prototypes.
>
> >Q3. It is unclear why an orthogonal mapping between the two spaces could keep the geometry, and also what geometry is kept here.
>
> A3.  The orthogonality constraint ensures that the mapping is \emph{distance-preserving (isometric)}. Specifically, for any two features $x_i, x_j$, we have $||x_i - x_j||_2 = ||W x_i - W x_j||_2$
> which implies that no scaling or distortion is introduced, and the intrinsic geometry of the VFM space is preserved.
>
> >Q4. Here, what data is used for learning this mapping is unclear at all.
>
> A4. The data used for learning the mapping corresponds to the \emph{training split} of each dataset, as described in the experimental section and supplementary material.
> Our setting follows the standard protocol in prior unsupervised adaptation works, ensuring a fair comparison.
>
> >Q5.  When discussing previous works in the 2nd paragraph, it is suggested that the specific reference is made so that readers can understand better.
>
> A5. We thank the reviewer for the suggestion. In fact, we have already included citations to representative methods in the second paragraph.
>
> >Q6. Novelty limitation，...
>
> A6. We agree that vision–language alignment has been extensively studied. However, most existing works rely on supervised learning with image–text pairs and require access to model parameters.
> In contrast, our work focuses on a different and less explored setting: \emph{unsupervised alignment without access to model parameters}.
> We have cited relevant works in the paper, and would greatly appreciate it if the reviewer could point us to additional closely related works that fit this setting. We will carefully study and include them in the revision.
>
> >Q7. This method needs to provide a set of text prompts which are assumed to be available.
>
> A7. We would like to clarify that providing a set of text prompts is a standard and minimal assumption in VLM tasks such as zero-shot classification and open-vocabulary segmentation.
> In these settings, the candidate category names (i.e., text prompts) define the label space. Without them, VLMs (e.g., CLIP) cannot perform classification or segmentation, as there is no semantic target to match against.
> Therefore, our method is \emph{unsupervised} in the sense that it does not require any labeled image–text pairs or annotations, and only relies on category names as in prior work.
>
> >Q8. What means by geometry structure in this context?
>
> A8. In our context, “geometry structure” refers to the \emph{relative relationships among visual features in the embedding space}, including pairwise distances, neighborhood relations, and cluster structure.
>
> >Q9. It is rather confusing that some test time adaptation methods (e.g., COSMIC, TIPPLE) is compared in Table 1. Is this work doing the same setting? It seems this work is messed up from problem, writing, all way to the evaluation setting.
>
> A9. Our setting is \emph{zero-shot classification without test-time adaptation}, as stated in Sec. 4.2. GPUA learns a mapping offline and directly applies it at inference time, without using test-time statistics or adaptation.
> Methods such as COSMIC and TIPPLE are included as strong baselines since they represent the state-of-the-art under related evaluation protocols. While they involve test-time adaptation, we include them to provide a broader comparison. Importantly, GPUA does not rely on test-time updates, yet achieves competitive or superior performance.
> We acknowledge that this distinction was not sufficiently emphasized and will clarify the evaluation setting and comparison rationale more explicitly in the revision to avoid confusion.

---

> > ### Author Rebuttal · Reviewer_G3hR · 2026-04-02
> >
> > Thanks for the clarification by the authors and these should be incorporated to improve the readability of this paper. I would adjust my score accordingly.

---

> > > ### Author Response · Authors · 2026-04-03
> > >
> > > Thank you for reading our rebuttal and updating your score. We will incorporate the relevant content into the revised manuscript to improve the readability of the paper.

---

### Decision · Program_Chairs · 2026-04-30

**Decision:**

Accept (regular)

**Comment:**

This paper proposes a framework, GPUA, to bridge the VFMs and VLMs. Before the rebuttal, the ratings are mixed. The authors made a great effort on the rebuttal; thus, most of the concerns from reviewers were addressed. All the reviewers appreciated the fact that the method is conducted in an unsupervised manner and achieves consistent improvements on zero-shot setting. The authors should revise their paper to improve its clarity according to the reviews. Hence, the AC recommends acceptance.